# Rethinking Optimal Transport in Offline Reinforcement Learning

**Arip Asadulaev**[1,2]    **Rostislav Korst**[3]    **Alexander Korotin**[4,1]    **Vage Egiazarian**[5,6]
**Andrey Filchenkov**[2]    **Evgeny Burnaev**[4,1]

[1]AIRI    [2]ITMO    [3]MIPT    [4]Skoltech    [5]Yandex    [6]HSE University

asadulaev@airi.net

## Abstract

We propose a novel algorithm for offline reinforcement learning using optimal transport. Typically, in offline reinforcement learning, the data is provided by various experts and some of them can be sub-optimal. To extract an efficient policy, it is necessary to *stitch* the best behaviors from the dataset. To address this problem, we rethink offline reinforcement learning as an optimal transport problem. And based on this, we present an algorithm that aims to find a policy that maps states to a *partial* distribution of the best expert actions for each given state. We evaluate the performance of our algorithm on continuous control problems from the D4RL suite and demonstrate improvements over existing methods.

## 1  Introduction

Deep reinforcement learning (RL) has shown remarkable progress in complex tasks such as strategy games [51], robotics [37], and dialogue systems [39]. In online RL, agents operate in the environment and receive rewards that are used to update policies. However, in some domains, such as healthcare, industrial control, and robotics interactions with the environment can be costly and risky. For these reasons, offline RL algorithms that use historical data of expert interactions with the environment are more applicable. Given only a dataset of experiences, offline RL algorithms learn a policy without any online actions [33].

Due to the off-policy nature of offline RL, naive training of online RL algorithms in offline settings does not provide effective solutions [33]. This is primarily due to a distribution shift between the learned and the behavior policy [15], which causes the instability of the critic function during off-policy evaluation. To solve this problem and make actions of the learned policy more similar to the one on which critic function was trained, a Behavior Cloning (BC) objectives was proposed [52, 33].

As an example of BC loss, Optimal Transport (OT) [50] distance has been introduced for RL. This application of OT is particularly relevant because by minimizing the OT distance between policies, we can efficiently clone the behavior [52, 11, 40, 34, 20, 9, 35]. But there is another problem, in offline RL, the dataset often consists of various experts demonstrations. Some of these demonstrations may be incorrect, or efficient only for certain parts of the environment. Consequently, cloning the behavior policy in such scenarios may limit policy improvement [30]. To solve these types of problems, RL algorithms need to apply *stitching*, which means that the algorithm selects the best action for each state and recombines different trajectories provided by multiple experts [30].

In our paper, we rethink OT as a framework for offline reinforcement learning that aims to stitch the best trajectories, rather than clone the policy. Unlike previous OT-based approaches in RL, we do not introduce a new OT-based regularization or a reward function [11, 40]. Instead, we propose **a**

**novel perspective that views the entire offline RL problem as an optimal transport problem** between state and actions distribution (§3). By utilizing the $Q$-function as a transport cost and the policy as an optimal transport map, we formalize offline reinforcement learning as a Maximin OT optimization problem. This opens doors for applying recent advantages of OT methods to RL directly. In particular, we propose an algorithm that trains a policy to identify the *partial* distribution of the best actions for each given state.

**Contribution**: We proposed a novel optimal transport-based policy extraction method and provided an analysis of its performance on various RL-based cost functions. The core of our method is the *Partial Policy Learning* (PPL) algorithm, which efficiently identifies the best action for each state in the dataset. We evaluated our method across various environments using the D4RL benchmark suite, achieving superior performance compared to state-of-the-art model-free offline RL techniques.

## 2 Background and Related Work

### 2.1 Optimal Transport

**Primal** form of the optimal transport problem was first proposed by Monge [50]. Suppose there are two probability distributions $\mu$ and $\nu$ over measurable spaces $\mathcal{X}$ and $\mathcal{Y}$ respectively, where $\mathcal{X}, \mathcal{Y} \subset \mathbb{R}^D$. We want to find a measurable map $T : \mathcal{X} \to \mathcal{Y}$ such that the mapped distribution is equal to the target $\nu$, for a cost function $c : \mathcal{X} \times \mathcal{Y} \to \mathbb{R}$, the *OT* problem between $\mu, \nu$:

$$\min_{T \sharp \mu = \nu} \mathbb{E}_{x \sim \mu} \big[ c(x, T(x)) \big]. \tag{1}$$

where $T\sharp$ is the push forward operator and respectively $T\sharp\mu = \nu$ represents the mass preserving condition. Informally, we can say that the cost is a measure of how hard it is to move a mass piece between points $x \in \mathcal{X}$ and $y \in \mathcal{Y}$ from distributions $\mu$ and $\nu$ correspondingly. That is, an OT map $T$ shows how to optimally move the mass of $\mu$ to $\nu$, i.e., with the minimal effort. A widely recognized alternative formulation for optimal transport was introduced by Kantorovich [22]. Unlike the Monge's OT problem formulation, this alternative allows for mass splitting. The Kantorovich OT problem can be written as:

$$\min_{g \in \Pi(\mu, \nu)} \mathbb{E}_{(x,y) \sim g} \big[ c(x, y) \big]. \tag{2}$$

In this case, the minimum is obtained over the transport plans $g$, which refers to the couplings $\Pi$ with the respective marginals being $\mu$ and $\nu$. The optimal $g^*$ belonging to $\Pi(\mu, \nu)$ is referred to as the *optimal transport plan*.

**Dual** form of optimal transport, following the Kantorovich-Rubinstein duality [49], can be obtained from (2) and written as:

$$\max_f \mathbb{E}_{x \sim \mu} \big[ f^c(x) \big] + \mathbb{E}_{y \sim \nu} \big[ f(y) \big], \tag{3}$$

where $f : \mathcal{X} \to \mathbb{R}$ and $f^c(x) = \min_{y \in \nu} \big[ c(x, y) - f(y) \big]$ is referred to as the $c$-transform of potential $f$ [50]. For cost function $c(x, y) = \|x - y\|_2$, the resulted OT cost is called the *Wasserstein-1* distance, see [50, §1] or [44, §1, 2]. It was shown that to compute *Wasserstein-1* distance, instead of computing conjugate function $f^c$ we can simply consider $f$ to be from the set of *1-Lipschitz* functions[50, 1].

**Maximin** formulation for simultaneously computing the OT distance and recovering the OT map $T$ was recently proposed. According to [27], by applying the Rockafellar interchange theorem [43, Theorem 3A], we can replace the optimization over points $y \in \nu$ with an optimization over functions $T : \mathcal{X} \to \mathcal{Y}$, which reformulates the problem (3) as a saddle-point optimization problem for the potential $f$ and map $T$:

$$\max_f \min_T \mathbb{E}_{x \sim \mu} \big[ c(x, T(x)) - f(T(x)) \big] + \mathbb{E}_{y \sim \nu} \big[ f(y) \big] \tag{4}$$

Using this formulation, significant progress has been achieved in utilizing neural networks for computing OT maps to address strong [36, 24, 18], weak [27], and general [2] OT problems. Our work is inspired by these developments, as we leverage neural optimal transport methods to improve offline reinforcement learning.

**Partial OT**: In cases where it is necessary to ignore some data and map the input to part of the target distribution, techniques like *unbalanced* or *partial* optimal transport are commonly employed [12, 4,

17]. For example, *partial* optimal transport is useful in resource allocation problems where only a subset of resources needs to be optimally allocated to match a subset of demands [12].

In our paper, we consider the partial OT formulation proposed by [17]. This framework, named *extremal OT*, solves the partial alignment between the full input distribution and part of the target distribution. Essentially, for *Euclidean* cost functions such as $\ell^2$, partial transport maps can be seen as a tool for finding *nearest neighbors* (maximally close) to the input samples from the target according to the cost function. Please see Figure 1 in [17]. Formally, this method can be written as:

$$\min_{T\sharp\mu\leq w\nu} \mathbb{E}_{x\sim\mu}\big[c(x,T(x))\big]. \tag{5}$$

We can call $w$ a coefficient of unbalance. The intuition behind this formulation is as follows: when $w=1$, the constraint $T\sharp\mu \leq w\nu$ is equivalent to $T\sharp\mu = \nu$. This equivalence holds because there is only one probability measure that is less than or equal to $\nu$, and it is $\nu$ itself. However, with values of $w \geq 1$, the mass of the second measure is scaled. Consequently, it is sufficient to match $\mu$ only to part of the second measure $\nu$ to transfer the full mass contained in $\mu$. Since the cost $c$ is our objective, $T$ will tend to map to the samples from $\nu$ that are closest to the input distribution. It was shown that this formulation can also be considered in maximin settings [17], which is the core of our method (§3.2).

## 2.2 Offline Reinforcement Learning

**Reinforcement Learning** is a well-established framework for decision-making processes in environments modeled as Markov Decision Processes (MDPs). The MDP is defined as $\mathcal{M} = (S, A, P, r, \gamma)$ by the state space $S$, action space $A$, transition probability $P(s' \mid s, a)$, reward function $r(s, a)$, and discount factor $\gamma$. The goal in RL is to find a policy $\pi(a|s)$ that maximizes the expected cumulative reward over time $t$: $J(\pi) \stackrel{def}{=} \mathbb{E}_\pi[\sum_{t=0}^{\infty} \gamma^t r(s_t, a_t)]$. Also we define the distribution over the state space following policy $\pi$ as $d^\pi(s)$. To estimate of the expected cumulative reward following a given policy $\pi$ the critic function $Q^\pi$ is used:

$$Q^\pi(s, a) = r(s, a) + \gamma\mathbb{E}_{s'\sim T(s,a), a'\sim\pi(s')}\left[Q^\pi(s', a'))\right]. \tag{6}$$

In deep RL neural networks are used to parameterize critic $Q^\pi_\phi(s, a)$ and policy $\pi_\theta$. Critic can be learned by minimizing the mean squared Bellman error over the experience replay dataset $\mathcal{D} = \{(s_i, a_i, s'_i, r_i)\}$, which consist of trajectories implied by policy $\pi$:

$$\min_{Q^\pi_\phi}\mathbb{E}_{(s,a,s')\sim\mathcal{D}}\left[\left(r(s, a) + \gamma\mathbb{E}_{a'\sim\pi_\theta(s')}\left[Q^\pi_\phi(s', a')\right] - Q^\pi_\phi(s, a)\right)^2\right]. \tag{7}$$

Using a trained critic function we can recover a near-optimal policy, for example, using Deterministic Policy Gradient (DPG) [45] or its improved version named Twin Delayed Deep Deterministic Policy Gradient (TD3) [15].

**Offline RL** is a fully data-driven approach for decision-making problems. Conventional offline RL algorithms use the (7) formula to recover the $Q$ function, using data $\mathcal{D}$ collected according to the expert policy $\beta$. However, due to *distribution shift* between actions from $\mathcal{D}$ and those induced by policy $\pi$, the $Q$ function may suffer from inefficiency and overestimation bias during evaluation [15]. To mitigate this issue, various offline RL algorithms with behavior cloning objectives was proposed. See the related work section (§6) for more details.

**OT in offline RL** has been proposed primarily for efficient behavior cloning. Compared to BC/KL, OT offers more flexibility, while including many building blocks to properly consider the geometry of the problem. The simplest method of integrating OT into offline RL is by utilizing the *Wasserstein-1* distance as the measure of dissimilarity between the learned and expert policies. This method is known as the W-BRAC algorithm [52]. Given the critic function $Q(s, a)$, the *Wasserstein-1* distance defined by $f$, and the behavior cloning coefficient $\alpha$, we have

$$\min_\pi \max_{\|f\|_L\leq 1} \mathbb{E}_{s\sim\mathcal{D}, a\sim\pi(s)}\big[\underbrace{-Q^\pi(s, a)}_{\text{Critic}}\big] + \alpha\big(\underbrace{\mathbb{E}_{(s,a)\sim\mathcal{D}}\big[f(s, a)\big] - \mathbb{E}_{s\sim\mathcal{D}, a\sim\pi(s)}\big[f(s, a)\big]}_{\text{Wasserstein-1 distance}}\big) \tag{8}$$

In real-world applications, collecting large and high-quality datasets may be too costly or impractical. Due to that, the provided data it is often collected by many expert playing several near-optimal policies and only a single expert can be optimal in dataset $\mathcal{D}$. The formulation (8) has no mechanism to infer the importance of each action, its just clone the entire data. Moreover, the problem is that computing *Wasserstein-1* distance is typically *complicated* [25], because the potential $f$ is required to satisfy the *1-Lipschitz* condition [1]. It is also important to note that the coefficient $\alpha$ is task-dependent[6].

---

**Algorithm 1** Partial Policy Learning

---

**Input:** Dataset $\mathcal{D}(s, a, r, s')$
Initialize $Q_\phi, \pi_\theta, f_\omega, \beta$
**for** $k$ in 1...N **do**
    $(s, a, r, s') \leftarrow \mathcal{D}$: sample a batch of transitions from the dataset.
    $Q^{k+1} \leftarrow$ Update cost function $Q_\phi^\pi$ using the Bellman update in (2).
    $f^{k+1} \leftarrow$ Update $f_\omega$ using outputs of $\pi_\theta$ and samples from dataset: $\arg\min_f -\mathbb{E}_{s\sim\mathcal{D}, a\sim\pi^k(s)}[f^k(s,a)] + w\mathbb{E}_{s,a\sim\mathcal{D}}[f^k(s,a)]$
    $\pi^{k+1} \leftarrow$ Update policy $\pi_\theta$ as a transport map: $\arg\min_\pi \mathbb{E}_{s\sim\mathcal{D}, a\sim\pi^k(s)}[-Q^k(s,a) - f^k(s,a)]$.
**end for**

---

## 3 Rethinking OT in RL

### 3.1 Considering RL as OT problem

To extend a connection and apply OT methods in RL, we can view the entire offline RL problem as an optimal transport problem. For this purpose, let's consider (1). By replacing the cost function $c(x,y)$ with the critic $Q^\pi(s,a)$ and treating our policy $\pi$ as a map that moves mass from the state distribution $d^\beta(s)$, ($d(s)$ for shorthand), to the corresponding distribution of actions given by the behavior policy $\beta(\cdot|s)$, we obtain a primal state-conditional Monge OT problem. $\mathbb{E}_{s\sim\mathcal{D}, a\sim\pi(s)}\min_{\pi\sharp d(s)=\beta(\cdot|s)}[-Q^\pi(s,a)]$ which is equal to:

$$\min_{\left\{\pi\sharp d(s)=\beta(\cdot|s)\right\}} \mathbb{E}_{s\sim\mathcal{D}, a\sim\pi(s)}\big[-Q^\pi(s,a)\big]. \tag{9}$$

The objective is to minimize the expectation of the negative critic function $Q^\pi$, (we can also consider $Q^\beta$) while mapping exclusively to the distribution of actions given by the expert policy $\beta$.

**Remark 1:** *In this formulation, the imposed equality constraints force the policy to mimic the behavior of the expert policy. However, since the provided dataset often contains sub-optimal paths, this formulation remains inappropriate for avoiding inefficient actions provided by the expert policy.*

In the following subsection, we introduce an algorithm that integrates Partial Optimal Transport [17] for deep RL.

### 3.2 *Stitching* with Partial OT

In many real-world applications, collecting high-quality datasets may be costly or impractical. Therefore, offline RL algorithms need to select the best actions provided by the experts and ignore sub-optimal ones. Our idea is that partial OT can help us deal with sub-optimal trajectories by mapping states only to the most relevant parts of the action distribution with respect to the critic function $Q$ To obtain partial formulation of 3, instead of equality constraints, we need to consider optimization problem with the following *inequality* constraints [17]:

$$\min_{\left\{\pi\sharp d(s)\leq w(\beta(\cdot|s))\right\}} \mathbb{E}_{s\sim\mathcal{D}, a\sim\pi(s)}\big[-Q^\pi(s,a)\big]. \tag{10}$$

As discussed in (§2.1), the parameter $w$ scales the mass contained in the target distribution. For values of $w \geq 1$, the mass of the second measure is increased. Consequently, it is not necessary to cover the full distribution $\beta(\cdot|s)$ to transport the Dirac mass given by $d(s)$. As a result, a learned policy (map) matches the state only to part of the action distribution $\beta(\cdot|s)$. This is particularly relevant for offline RL, as the resulting policy will not clone all actions but will select only the most optimal ones, where optimality is defined by the critic function $Q$.

**Proposition 1:** *(Informal) for a critic-based cost function $Q^\beta$, an offline Policy trained by* (10) *improves over the behavior policy $\beta$.* Proofs are given an Appendix (§9.1).

To find a practical solution to the given problem using neural networks, we can also consider a dual and maximin formulation for (10). The dual form for the partial transport problem[17] can be expressed as

$$\max_{f\geq 0} \mathbb{E}_{s\sim\mathcal{D}, a\sim\pi}\big[f^c(s,a)\big] + w\mathbb{E}_{s\sim\mathcal{D}, a\sim\beta}\big[f(s,a)\big]. \tag{11}$$

To obtain the maximin formulation, we expand the c-transform (3) using the function $Q$ as the cost $c$. This operation can be represented as $\mathbb{E}_{s \sim \mathcal{D}, a \sim \pi}[f^c(s,a)] = \mathbb{E}_{s \sim \mathcal{D}}[\min_a \{-Q^\pi(s,a) - f(s,a)\}]$. Subsequently, by applying the Rockafellar interchange theorem [43, Theorem 3A], we replace the optimization over points $a$ with an equivalent optimization over functions $\pi : \mathcal{S} \to \mathcal{A}$. The solution can be computed using neural approximations, which is useful for RL tasks

$$\max_{f \geq 0} \min_{\pi} \mathbb{E}_{s \sim \mathcal{D}, a \sim \pi(s)} \Big[ \underbrace{-Q^\pi(s,a) - f(s,a)}_{\text{Cost}} + \underbrace{w \mathbb{E}_{(s,a) \sim \mathcal{D}} \big[f(s,a)\big]}_{\text{Constraints}} \Big]. \tag{12}$$

**Remark 2:** *In the [17, Proposition 4] it was shown that $f^*(s,a) = 0$ vanishes on the outliers i.e on the samples outside the scaled target distribution. The policy update in (12), can be written as $\min_\pi \mathbb{E}_{s \sim \mathcal{D}, a \sim \pi(s)}[-Q(s,a) - f(s,a)]$. This reveals that if $f(s,a) = 0$ exclusively on the outlier actions, it then applies extra weight to correct actions, which reduces the value of $-Q(s,a) - f(s,a)$.*

**Remark 3:** *The value of $w$ controls the size of the action space in which we map; the higher the value of $w$, the smaller the subset[17]. When the actions are provided by a single expert, scaling $w$ can be detrimental to performance.*

The primary distinction between our method (12), and previous OT for RL approaches, particularly (8), is as follows:

- OT is not treated as a component of the problem or as a regularization term with a coefficient of $\alpha$. Instead, the entire policy optimization is considered as OT.

- Our formulation for computing OT does not necessitate *1-Lipschitz* function constraints on $f$, as it does not compute the *Wasserstein-1* distance. Rather, it addresses a maxmin OT problem with a critic-based cost function, where $f$ can be any arbitrary scalar function satisfying $f \geq 0$.

- Unlike traditional OT approaches that fully aligns two distributions, our method maps only to the part of the target distribution with the best actions with respect to the critic.

In practice, we use neural networks $\pi_\theta : S \to A$ and $f_\omega : S \times A \to \mathbb{R}$ to parameterize $\pi$ and $f$ respectively. These neural networks serve as approximators that can capture complex mappings of states. We train these neural networks using stochastic gradient ascent-descent (SGAD) by sampling random batches from the dataset $\mathcal{D}$. At each training step, we sample a batch of transitions from the offline dataset $\mathcal{D}$ and then adjust the value function $Q$, and the potential $f$, Then for the $k$ update step, given the last policy value function $Q^k$, and $f^k$, we update the policy, see Algorithm 1.

## 4 Toy Experiments

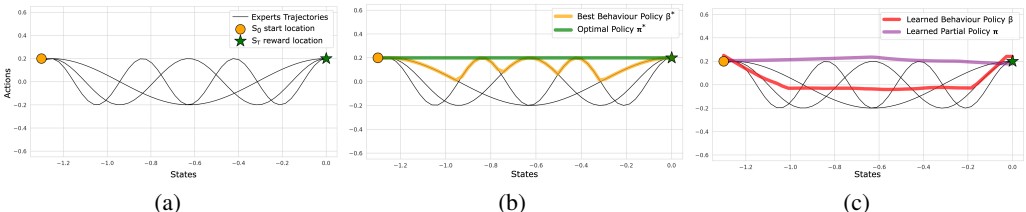

(a)                      (b)                      (c)

Figure 1: Toy experiments. (a) Left point $S_0$ denoting start and $S_T$ is the only rewarded, target location. Black curves visualize behavior trajectories $\beta$. (b) Best behavior policy $\beta^*$ according to the data, and the optimal policy $\pi^*$ that provides the optimal (shortest path) solution. (c) Results of the policy $\beta$ trained by minimizing $-Q^\beta(s, \pi(s))$+BC and policy $\pi$ trained by our algorithm 1.

To demonstrate the ability of our algorithm to extract the best policy from the sub-optimal behavior trajectories, we constructed a simplified offline RL dataset. The goal is to find the shortest path from a starting location $S_0$ to a reward point $S_T$, navigating through 50 intermediate states $S_t$ uniformly spaced along the $x$-axis from -1.3 to 0. The action space is continuous, with the agent's task being to determine the optimal $y$-values for the each $x$-coordinate. For the each intermediate state-action pair, the reward is defined as negative *Euclidean distance* to the final state-action pair $S_T, A_T$.

The offline RL dataset consisted of three sub-optimal expert trajectories, as shown in Figure 1(a). The best policy that can be extracted by recombining the expert trajectories and an actual optimal policy $\pi^*$ that provides the shortest path are highlighted in Figure 1(b).

We compared our method with the offline RL algorithm based on behavior cloning. For this, we trained the critic $Q^\beta$ by (2), using actions drawn from the observed behavior trajectories $\beta$ presented in the dataset $\mathcal{D}$. Given $Q^\beta$, we minimize $\mathbb{E}_{s,a\sim\mathcal{D}}[-Q^\beta(s,\beta(s)) + \|\beta(s) - a\|^2]$ to obtain the policy $\beta$.[1] Next, using the given critic function $Q^\beta$, we trained the policy $\pi(s)$ using our algorithm with parameter $w = 8$. Compared to offline RL 1(b), which produced *average expert* solutions, our method demonstrated superior performance by extracting and exploiting the most strategic insights from the data, maximizing the reward and ignoring all inefficient actions, see Figure 1(c)..

In all experiments, a single-layer neural network with 32 neurons and ReLU activation was used as the configuration for both the critic and policy networks. For our method, we used a Lagrangian network $f(s,a)$ with similar parameters. There are 5000 steps was done for each network, with Adam [23] optimizer with a learning rate (lr) of 1e-3.

## 5 D4RL Experiments

### 5.1 Benchmarks and Baselines

We evaluate our proposed method using the Datasets for Deep Data-Driven Reinforcement Learning (D4RL) [13] benchmark suite, which is a collection of diverse datasets designed for training and evaluating deep RL agents in a variety of settings. It includes tasks in continuous control, discrete control, and multi-agent settings with a variety of reward structures and observation spaces. First, we tested our method on the Gym's `MuJoCo-v2` environments, such as Walker, Hopper, and HalfCheetah. We also tested our method on the complex `Antmaze-v2` and `Android-v1` environments.

To compare the performance of our proposed method, we selected four state-of-the-art offline RL algorithms. These include Conservative $Q$-Learning (CQL) [32], Twin Delayed Deep Deterministic Policy Gradient with behavior Cloning (TD3+BC) [14], Implicit $Q$-Learning IQL [28], ReBRAC [47], and IQL with Optimal Transport Reward Labeling (IQL+OTR) [35]. More details on the methods are given in (§6). The results of the supervised behavior cloning are also included.

### 5.2 Settings

We tested our method in conjunction with various algorithms that avoid overestimation of Q-function. These overestimation-avoiding methods are used to obtain different types of *cost functions* in our method, showing that our method can work efficiently with any of them.

**PPL**. First, to be consistent with Proposition 1, we considered our method in the One-Step RL settings. Following the original setting provided by One-Step RL [6], we pre-trained $\beta$ for 500k steps. Next, we pre-trained $Q^\beta$ for 2 million steps. Then, to avoid overestimation bias, we applied a *simplified* conservative update to $Q^\beta$ (see [32, Eq.1]). Finally, to improve beyond the behavior policy $\beta$, we trained the policy $\pi$ for the 100k steps using our method 1. For these experiments, a two-layer feed-forward network with a hidden layer size of 1024 and a learning rate of 0.001 was used with the Adam optimizer [23]. We tested this approach for the `Mujoco` environment see Table 4, for the comparison with the OneStep RL method.

**PPL$^{\text{CQL}}$**. Second, we tested our method in conjunction with the CQL method on the `Antmaze` problems. In this setting, no pre-trained models were used; we used the code provided by the CORL library [48], with all hyperparameters and architectures set identical to those originally used in CORL for the CQL method. We trained the algorithm for 1M steps, with $w$ set to 8 for all experiments. See the comparison with CQL in Table 1.

**PPL$^{\text{R}}$**. Finally, we coupled our method with the improved version of TD3+BC called ReBRAC [47]. In these experiments, for a cost function, the TD3+BC objective was used, which combines the $Q$ function with a supervised loss. The same hyperparameters proposed by the authors for ReBRAC were kept in our experiments. No pre-trained models were used, we trained the agent from scratch for

---

[1]We also tested direct minimization of $-Q^\beta(s,a)$. However, even in toy settings this method suffered from overestimation bias, causing the resulting policy to consistently predict values equal to 1.

1M steps. We have tested our method on the `Mujoco`, `Android`, and `Antmaze` datasets. While the ReBRAC framework recommends task-specific hyperparameters [47], the results for the `Antmaze` are also reported for the best $w$ hyperparameter. The results can be found in the Tables 1, 2 and 3.

## 5.3 Results

In the Table 1 we consider a side-by-side comparison, and the color purple indicates the best results between two columns. In the Tables 2 and 3 the color indicates the top 3 results. We compared the results with those reported in the ReBRAC [47] and OTR+IQL papers [35]. For the `Antmaze` datasets, the main comparison is with the *oracle* method to which our method is coupled. For comparison, we reproduced the CQL and ReBRAC results using the code provided in CORL. The CQL score curves are provided in Appendix 3. For completeness, IQL and OTR+IQL results are also included.

Table 1: Averaged normalized scores on `Antmaze-v2` tasks. Reported scores are the results of the final 100 evaluations and 5 random seeds.

| Dataset | IQL | OTR+IQL | CQL | $PPL^{CQL}$(Ours) | ReBRAC | $PPL^{R}$(Ours) |
|---|---|---|---|---|---|---|
| umaze | 87.5 ± 2.6 | 83.4 ± 3.3 | 86.3 ±3.7 | 90±2.6 | 97.8 ± 1.0 | 98.0 ±14 |
| umaze-diverse | 62.2 ± 13.8 | 68.9 ± 13.6 | 34.6 ±20.9 | 40±2.6 | 88.3 ± 13.0 | 93.6±6.1 |
| medium-play | 71.2 ± 7.3 | 70.5 ± 6.6 | 63.0 ±9.8 | 67.3±10.1 | 84.0 ± 4.2 | 90.2 ±3.1 |
| medium-diverse | 70.0 ± 10.9 | 70.4 ± 4.8 | 59.6 ±3.5 | 65.3±8.0 | 76.3 ± 13.5 | 84.8 ±14.7 |
| large-play | 39.6 ± 5.8 | 45.3 ± 6.9 | 20.0 ±10.8 | 25.6±3.7 | 60.4 ± 26.1 | 76.8 ±4.0 |
| large-diverse | 47.5 ± 9.5 | 45.5 ± 6.2 | 20.0 ±5.1 | 23.6 ±11.0 | 54.4 ± 25.1 | 76.6 ± 7.4 |
| Total | 378 | 384 | 283.5 | 311.8 | 461.2 | **520** |

Table 2: Averaged normalized scores on `MuJoCo` tasks. Reported scores are the results of the final 10 evaluations and 5 random seeds.

| | Dataset | BC | One-RL | CQL | IQL | OTR+IQL | TD3+BC | ReBRAC | $PPL^{R}$(Ours) |
|---|---|---|---|---|---|---|---|---|---|
| M | Half. | 42.6 | 48.4 | 44.0 | 47.4 | 43.3 | 48.3 | 65.6 | 64.95±0.2 |
| | Hopper | 52.9 | 59.6 | 58.5 | 66.3 | 78.7 | 59.3 | 102.0 | 93.49±7.2 |
| | Walker | 75.3 | 81.1 | 72.5 | 78.3 | 79.4 | 65.5 | 82.5 | 85.66±0.6 |
| MR | Half. | 36.6 | 38.1 | 45.5 | 44.2 | 41.3 | 44.6 | 51.0 | 51.1±0.3 |
| | Hopper | 18.1 | 97.5 | 95.0 | 94.7 | 84.8 | 60.9 | 98.1 | 100.0±2 |
| | Walker | 26.0 | 49.5 | 77.3 | 73.9 | 66.0 | 81.8 | 77.3 | 78.66±2.0 |
| ME | Half. | 55.2 | 93.4 | 91.6 | 86.7 | 89.6 | 90.7 | 101.1 | 104.85±0.1 |
| | Hopper | 52.5 | 103.3 | 105.4 | 91.5 | 93.2 | 98.0 | 107.0 | 109.0±1.2 |
| | Walker | 107.5 | 113.0 | 108.8 | 109.6 | 109.3 | 110.1 | 112.3 | 111.74±1.1 |
| | Total | 467.7 | 684.9 | 698.6 | 692.6 | 685.6 | 659.2 | 796.9 | **799.45** |

In offline RL, the problems are noisy, complex, and diverse, and task-specific hyperparameter search is required. Despite this fact, our method shows promising results and consistently outperforms methods on which it is based. Most importantly our method gives significant improvement in the achieved scores for the `Antmaze` problems (Table 1). In particular, our method provides state-of-the-art results for all datasets on this task, and gives a significant improvement of (+16) and (+21) points for the complicated large-play-v2 and large-play-diverse-v2 environments. Importantly, we consistently outperform the previous best OT-based offline RL algorithm, OTR+IQL.

We can interpret that *our method lies between behavior cloning and direct maximization of the Q function.* Recent studies have shown that direct maximization can lead to sub-optimal results due to overestimation bias [16]. Conversely, being too close to the expert's policy prevents improvement [30]. Intuitively, our extremal formulation allows us to strike a balance, maximizing $Q$ by actions from the expert action distribution that are not radically different from those on which the $Q$ was trained.

Our formulation aligns the state space with only a portion of the action space. This property is particularly relevant for offline datasets that contain a *mixture* of expert demonstrations. Our method can help to select the best possible action from a set of expert trajectories [33]. In other words, optimal $f$ is to encourage the discovery of new actions that can yield high rewards. To illustrate the task-specific dependence of the parameter $w$, we performed an ablation for different values (§5.4).

Table 3: Averaged normalized scores on `Android` tasks. Reported scores are the results of the final 10 evaluations and 5 random seeds.

|  | Dataset | BC | TD3+BC | CQL | IQL | OTR+IQL | ReBRAC | PPL$^R$(Ours) |
|---|---|---|---|---|---|---|---|---|
| Pen | Human | 34.4 | 81.8 | 37.5 | 81.5 | 66.82 | 103.5 | 108.43 |
|  | Cloned | 56.9 | 61.4 | 39.2 | 46.8 | 78.7 | 91.8 | 113.97 |
|  | Expert | 85.1 | 146.0 | 107.0 | 133.6 | - | 154.1 | 152.33 |
| Door | Human | 0.5 | -0.1 | 9.9 | 3.1 | 5.9 | 0.0 | 1.0 |
|  | Cloned | -0.1 | 0.1 | 0.4 | 0.8 | 0.01 | 1.1 | 0.44 |
|  | Expert | 34.9 | 86.4 | 101.5 | 105.3 | - | 104.6 | 104.66 |
| Hammer | Human | 1.5 | 0.4 | 4.4 | 2.5 | 1.79 | 0.2 | 2.0 |
|  | Cloned | 0.8 | 0.5 | 2.1 | 1.1 | 0.8 | 6.7 | 4.74 |
|  | Expert | 125.6 | 117.0 | 86.7 | 106.5 | - | 133.8 | 130.2 |
| Relocate | Human | 0.0 | -0.2 | 0.2 | 0.1 | -0.2 | 0.0 | 0.23 |
|  | Cloned | -0.1 | -0.1 | -0.1 | 0.2 | 0.1 | 0.9 | 0.45 |
|  | Expert | 101.3 | 107.3 | 95.0 | 106.5 | - | 106.6 | 92.99 |
|  | Total | 340.9 | 600.5 | 484.8 | 588.5 | - | 703.2 | **711.44** |

**Run time:** The code is implemented in the `PyTorch` [41] and `JAX` frameworks and will be publicly available along with the trained networks. Our method converges within 2–3 hours on Nvidia 1080 (12 GB) GPU. We used WanDB [5] for babysitting training process. The code is available in supplementary materials.

## 5.4   Parameter $w$

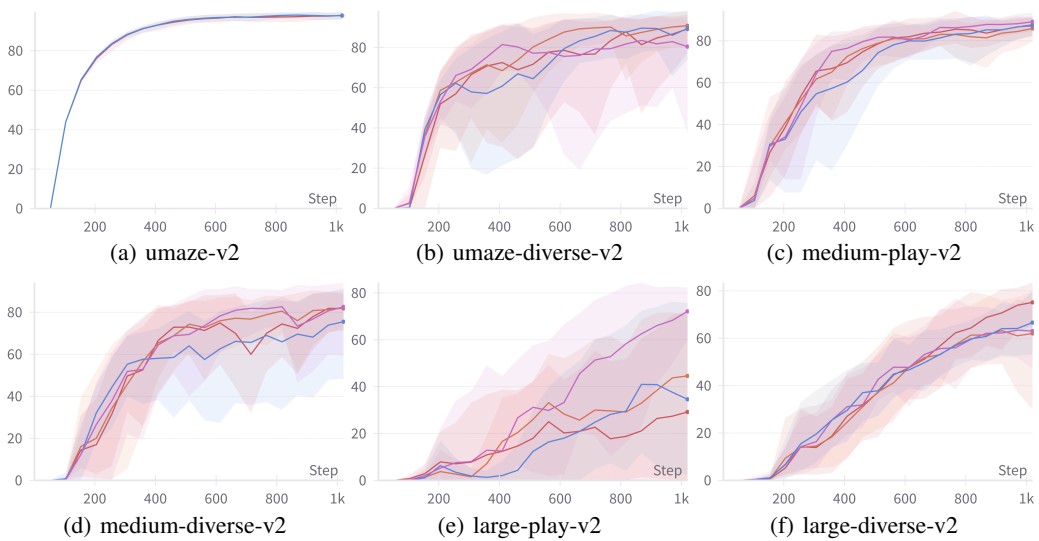

Figure 2: Exponential moving average (coef. 0.3) curves of the normalized score curves for the `Antmaze`. Different colors of the curves represent results for $w = 3$, $w = 5$, $w = 8$, $w = 12$.

The parameter $w$ influences the action selection process by determining the support range over which the policy operates. With the parameter $w$ equal to 1, our method considers all actions provided in the dataset for each state. By increasing this value, we consider the smaller subspace of possible actions, which can be useful when it is necessary to select the best one from the data. In a perfect scenario, we want to find as few actions as possible that maximize the score, so we favor the higher values of $w$.

For some datasets, reducing the action subspace is unnecessary. We studied different parameters $w$=[3,5,8,12] for the range of tasks, results for the `Antmaze` are presented in Figure 2. We can see that the biggest effect of the parameter was accrued for the large-play and large-diverse tasks 2 (e). This means that selecting the subspace of actions is important for this environments, and our method allows for this nuanced control depending on the task.

# 6 Related work

There is a rich family of offline RL algorithms. To solve the distribution shift problem, several methods have been proposed [6, 29, 53, 7, 31, 14, 28]. The simplest approach is the behavior regularized actor-critic (BRAC) [52]. Minimizing some divergence $D$, between the learned $\pi$ and the expert policy $\beta$, the BRAC framework can be written as *distance regularized* policy optimization: Where $D(\pi(\cdot|s), \beta(\cdot|s))$ is BC loss. Based on BRAC framework [14] proposed a TD3+BC approach that combines standard TD3 [15] with BC loss minimization between policy and expert actions, and showed strong performance.

Policy Gradient from Arbitrary Experience via DICE [38] proposes to incorporate $f$-divergence regularization into the critic function. Based on a similar concept, Conservative $Q$-Learning [32] and Adversarial Trained Actor-Critic [8] extends the critic loss with *pessimistic* terms that minimize the $Q$ values on the samples of a learned policy and maximize the values of the dataset actions.

Methods such as one-step RL [6], solves stitching by approximating the behavior $Q^\beta$ function and extract the corresponding policy $\pi$ by weighting actions using the advantage function $A^\beta(s, a)$ to find the best action for the given state [46]. Based on this concept, Implicit $Q$-Learning (IQL) [28] approximates the policy improvement step by treating the state value function $V^\beta(s)$ as a random variable and taking a state-conditional upper expectile to estimate the value of the best actions.

Application of optimal transport in offline RL was wiedly explored for behavioral cloning. For example it was used to construct a *pseudo-reward* function [11, 40, 34, 20, 9]. Primal Wasserstein Imitation Learning [11] and Sinkhorn Imitation Learning [40] use the OT distance between the expert and imitator to create a reward function. Recently, the Optimal Transport Reward Labeling (OTR) method [35] was proposed to generate a reward-annotated dataset that can then be used by various offline RL algorithms. However, like W-BRAC, these methods primarily use OT as an additional *loss*.

# 7 Conclusion and Discussion

In this paper, we have established the novel algorithm for offline reinforcement learning. Our work introduces the concept using the action-value function and the policy as components of the OT problem between states and actions. To demonstrate the potential of our formulation, we propose the practical algorithm that penalize the policy to avoid inefficient actions provided in the dataset. By applying partial optimal transport, our algorithm effectively selects and maps the best expert actions for each given state, ensuring efficient policy extraction from noisy or sub-optimal datasets. The *limitation* of our method is that parameter $w$ is task dependent.

This work is a step towards our larger vision towards deepening the connection between OT and offline RL RL. Using our formulation other OT methods also can be integrated into RL. For example, various regularizations [26, 10] and general costs [42, 2] can be used to incorporate task-specific information into the learned map, which can be particularly relevant in hierarchical RL problems [3]. Additionally Weak Neural OT [27] can be relevant in RL where stochastic behavior is preferred for exploration in the presence of multimodal goals [19].

## 7.1 Reproducibility

To reproduce our experiment we provide source code in `https://github.com/machinestein/PPL/`. Details on used hyperparameters are presented in settings (§5.2).

## 7.2 Societal Impact

This paper presents research aimed at advancing the field of RL. While our work may have various societal implications, we do not believe any particular ones need to be specifically emphasized here.

# 8 Acknowledgment

The work was supported by the Analytical center under the RF Government (subsidy agreement 000000D730321P5Q0002, Grant No. 70-2021-00145 02.11.2021)

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

# 9 Appendix

## 9.1 Proofs

**Proposition 9.1** (Policy Improvement with Partial Policy Learning). *For any policy $\pi$, let's define its performance as $J(\pi) \stackrel{def}{=} \mathbb{E}_\pi[\sum_{t=0}^\infty \gamma^t r(s_t, a_t)]$ over the trajectories obtained by following the policy $\pi$, $(s_0 \sim S, a_t \sim \pi(s_t), s_{t+1} \sim P(\cdot \mid s_t, a_t))$. Let $\beta$ be the policy of an expert and $\pi$ is the solution to Eq. 10 with the $Q^\beta$ cost function. Then it holds that:*

$$J(\pi) \geq J(\beta).$$

**Proof.** According to [21], to compare the performance of any two policies $\pi$ and $\beta$ we can use the *Performance difference lemma*:

$$J(\pi) - J(\beta) = \frac{1}{1-\gamma} \mathbb{E}_{s \sim d^\pi} \left[ A^\beta(s, \pi) \right] \tag{13}$$

By applying [17, Theorem 3], the solution of (10) as the parameter $w \to \infty$, converges to the solution of the following problem:

$$\min_{\left\{ \pi(\cdot|s) \subset \text{Supp}(\beta(\cdot|s)) \right\}} \mathbb{E}_{s \sim \mathcal{D}, a \sim \pi}[-Q(s, a)]. \tag{14}$$

Here support (Supp) is the set of points where the probability mass of the distribution lives. In our settings $\text{Supp}(\beta(\cdot|s))$ indicates the best action from $\beta$ which maximizes $Q$.

Now, to show the improvement over the behavior policy, let's consider the difference between the behavior policy $\beta$ and the new policy $\pi$ obtained after the update of $\beta$ using our Eq.14, then we have:

$$J(\pi) - J(\beta) = \frac{1}{1-\gamma} \mathbb{E}_{s \sim d^\pi} \left[ Q^\beta(s, \pi) - V^\beta(s) \right] = \tag{15}$$

$$\frac{1}{1-\gamma} \mathbb{E}_{s \sim d^\pi} [Q^\beta(s, \max_{a \subset \text{Supp}(\beta(\cdot|s))} [Q^\beta(s, a)]) - \mathbb{E}_{a \sim \beta(s)}[Q^\beta(s, a)]] \geq 0 \tag{16}$$

Where in first we used the definition of $\pi$, and then noted that the maximum over the all actions given by the expert is always greater than the average over some actions from the experts policy $\beta$. $\square$

This result is analogous to the classic policy iteration improvements proof. The difference is that we takes max over the $\text{Supp}(\beta(\cdot|s))$ rather than max over the all possible actions $A$.

## 9.2 Additional Illustrations

For completeness, we provide an additional illustration of the results of the proposed method. Specifically, we show the scores for the one-step settings Table 4 and in this subsection we show the normalized score curves of the CQL and our method during training Figure 3.

Table 4: Averaged normalized scores on `MuJoCo` tasks. Results of the final 10 evaluations and 5 random seeds.

|  | Dataset | OneStep-RL | PPL |
|---|---|---|---|
| Medium | HalfCheetah | 48.4 | 51.4±0.2 |
|  | Hopper | 59.6 | 80.4±7.4 |
|  | Walker | 81.1 | 84.3±0.6 |
| Medium-Replay | HalfCheetah | 38.1 | 44.8±0.5 |
|  | Hopper | 97.5 | 92.1±10.8 |
|  | Walker | 49.5 | 86.6±4.8 |
| Medium-Expert | HalfCheetah | 93.4 | 74.4±17.0 |
|  | Hopper | 103.3 | 110.2±1.9 |
|  | Walker | 113.0 | 111.2±0.7 |

**Parameters settings** As mentioned, we considered different values of $w$ for the ReBRAC-based method. The parameters can be seen in the Table 5. For most tasks, higher values of $w$ are preferable, which means that some sort of suboptimal trajectories are present in the datasets.

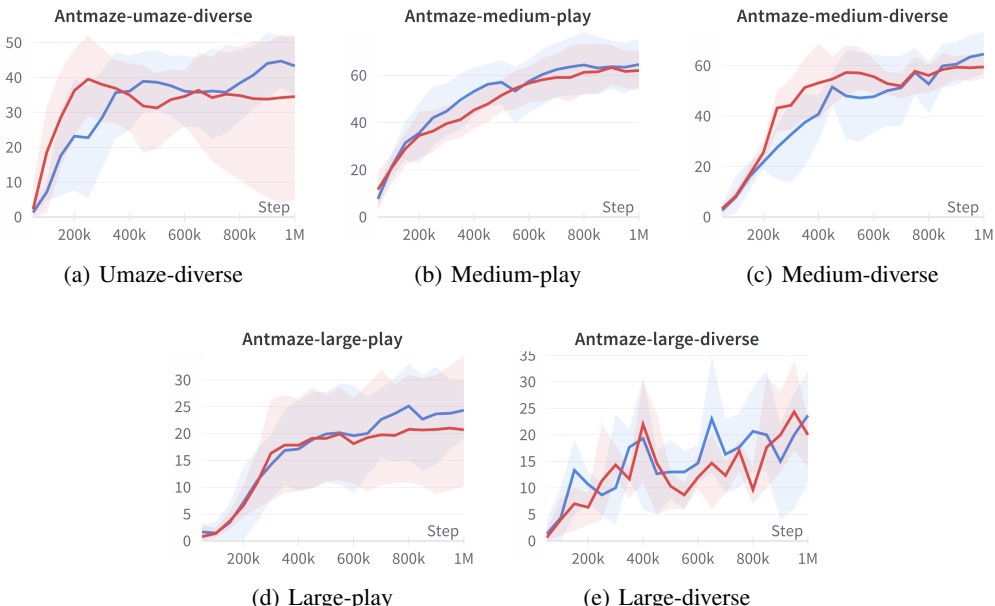

Figure 3: Normalized score curves on the Antmaze tasks, IPLCQL algorithm is blue, CQL is red

| $w$ | Environments |
|---|---|
| 8 | hopper-medium, pen-cloned, relocate-cloned, relocate-expert, antmaze-large-diverse, halfcheetah-medium, walker-medium, halfcheetah-medium-replay, hopper-medium-replay, walker-medium-replay, walker-medium-expert, umaze, door-cloned, door-expert, hammer-cloned |
| 12 | hopper-medium, pen-cloned, relocate-cloned, relocate-expert, antmaze-large-diverse |
| 5 | antmaze-large-play, antmaze-medium-play, antmaze-medium-diverse, pen-expert, hammer-expert |
| 3 | umaze-diverse, pen-human, relocate-human |

Table 5: Optimal $w$ values for different environments

