# OpenReview forum: "Rethinking Optimal Transport in Offline Reinforcement Learning"
_NeurIPS.cc/2024/Conference — NeurIPS 2024 poster_

### Official Review · Reviewer_7ePM · 2024-07-02

**Soundness:** 3
**Presentation:** 3
**Contribution:** 3
**Rating:** 7
**Confidence:** 4

**Summary:**

This paper provided a novel view of offline RL, which roughly is maximizing return while keeping close to the data, as OT problem. The key contribution is demonstrating that partial OT can effectively address the challenge of stitching—a fundamental issue in offline RL—both theoretically and empirically.

**Strengths:**

1. Formulation of offline RL as OT from stationary dist for state to datasets dist by some policy is promising.
2. The formalisation as OT is made convincing b y presenting the capability of problem of stitching by partial OT in principled way (hyperparameter w controls the degree of the algorithm between BC and naive value maximization)
3. The main claim—that partial OT effectively handles stitching—is supported by experimental results, particularly in maze environments known to require stitching.

**Weaknesses:**

In your method, stitching is done by the dynamic programing, application of bellman operator and your contribution regarding stitching is introducing partial OT to allow it work by alleviating the matching restriction. And, in high-level, it is what existing regularization methods like TD3+BC(BC regularization) or ReBRAC(KL regularization) are doing. While this is not a weakness, the paper could be strengthened by providing a clearer explanation of the advantages of OT compared to BC or KL regularization, which is partially addressed in the experiment section.

**Questions:**

Could you explain the theoretical/intuitive benefit of OT in offline RL compared with other technique to make policy close to the data such as behavior cloning or KL regularization?

**Limitations:**

Yes

---

> ### Author Rebuttal · Authors · 2024-08-07
>
> Thank you for your review and for a positive assessment of the paper! Your valuable feedback will help us improve the manuscript! Please find below the answers to your questions.
>
> **Q1: In your method, stitching is done by the dynamic programing, application of bellman operator and your contribution regarding stitching is introducing partial OT to allow it work by alleviating the matching restriction. And, in high-level, it is what existing regularization methods like TD3+BC(BC regularization) or ReBRAC(KL regularization) are doing. While this is not a weakness, the paper could be strengthened by providing a clearer explanation of the advantages of OT compared to BC or KL regularization, which is partially addressed in the experiment section. Could you explain the theoretical/intuitive benefit of OT in offline RL compared with other technique to make policy close to the data such as behavior cloning or KL regularization?**
>
> Compared to BC/KL, OT offers more flexibility, while including many building blocks to properly consider the geometry of the problem. For example
>
>
> - BC/KL regularization match the policy to the data distribution at a pointwise level, which can lead to suboptimal policy learning in scenarios where exact matching is not feasible or necessary. In OT, however, different constraints and regularizations can be used. The partial alignment proposed in our paper is a clear example.
> - OT methods allow the choice of arbitrary cost functions that take into account the domain of the problem. For example, we proposed to use Q-function-based cost function for transformation in RL domain. In comparison, BC/KL do not inherently consider the transformation costs between distributions.
>
> From a more general perspective, considering RL as OT bridges the gap between these two areas, allowing tools from OT to be applied in RL for better efficiency.  We will explicitly include this discussion in the final version of the paper.
>
>
> **Concluding remarks:** We truly value your reviews. We hope that clarifications are helpful. If there are remaining issues or questions on your mind, we're more than willing to address them.

---

> > ### Comment · Reviewer_7ePM · 2024-08-11
> > **Thank you for your detailed response**
> >
> > Thank you for your replay, the claim that by formulating as OT, we can leverage well-developed tool in OT in RL sounds nice. Looking forward to your future work showing their effective usage in RL.

---

> ### Comment · Area_Chair_ZGRe · 2024-08-11
> **Please respond to the authors**
>
> Hello reviewer 7ePM: The authors have responded to your comments. I would expect you to respond in kind.

---

### Official Review · Reviewer_iRBG · 2024-07-11

**Soundness:** 2
**Presentation:** 2
**Contribution:** 4
**Rating:** 6
**Confidence:** 4

**Summary:**

The authors propose a novel perspective for offline reinforcement learning by formulating offline reinforcement learning as a partial optimal transport problem. They view the policy $\pi$ as a transport map from the state distribution $d^\beta$ to $\beta(\cdot\mid s)$ and show that the dual form of the partial optimal transport problem can be expressed as a maximin optimization problem. The resulting maximin problem can be easily optimized, unlike other optimal transport problems, due to the absence of 1-Lipschitz constraints. Experimental analyses on various offline RL benchmarks manifest the effectiveness of the proposed algorithm, especially in the antmaze environments.

**Strengths:**

The paper introduces an interesting perspective of formulating offline RL as a partial optimal transport problem. Also, the proposed algorithm outperforms existing baselines on antmaze tasks by a huge margin.

**Weaknesses:**

1. The authors formulate offline RL as a partial optimal transport problem between $d^\beta$ and $\beta(\cdot\mid s)$ by regarding the policy $\pi$ as a transport map. Since the problem depends on the choice of $s$ in $\beta(\cdot\mid s)$, the resulting policy $\pi$ will also depend on it. This does not make much sense.

2. The explanation of the environment used in the toy example is unclear. Line 193 states that the final state yields a reward of 1, while other states have a zero reward. Then, what would necessitate the agent to seek the shortest path? The reward seems independent of the agent's action.

3. It is difficult to understand how PPL, PPL-CQL, and PPL-R work. A pseudocode for each variant would be helpful.

4. The paper has a huge room for improvement in terms of formatting. Excessive use of underlines, underfull hbox on Line 6 of Algorithm 1, and misaligned $\pm$s and ragged purple boxes in the tables hurt the paper's readability.

**Questions:**

Have you tried using a pre-trained Q-function learned by IQL or any other in-sample value learning algorithms instead of training the Q-function in parallel?

**Limitations:**

The authors adequately addressed the limitations of their work.

---

> ### Author Rebuttal · Authors · 2024-08-07
>
> Thank you for taking the time to review our paper and provide useful feedback. Your questions will help us improve the manuscript. Below are the answers to your questions.  Please let us know if any issues remain!
>
> **Q1: The authors formulate offline RL as a partial optimal transport problem between  and  by regarding the policy  as a transport map. Since the problem depends on the choice of $s$ in $\beta(\cdot|s)$, the resulting policy will also depend on it. This does not make much sense.**
>
> Please note that our method is completely offline, with no access to the online environment. The distribution over the states provided by the expert policy is everything we have in offline learning. All we have to learn from are the states visited by the expert policy $\beta$. The goal of our method is to extract the best policy using the given distribution of expert states and actions $\beta(\cdot|s)$. The dependence on $s$ in $\beta(\cdot|s)$ is the standard approach for offline RL, not something we invented (lines 16-17, 107-112).
> Please let us know if this answer addresses your concern.
>
>
> **Q2: The explanation of the environment used in the toy example is unclear. Line 193 states that the final state yields a reward of 1, while other states have a zero reward. Then, what would necessitate the agent to seek the shortest path? The reward seems independent of the agent's action.?**
>
> Our method works in the most common MDP settings with a discount factor of $\gamma < 1$ (lines 94-106), which prioritize immediate rewards over distant future rewards. By taking the shortest path to a reward, the agent ensures that it receives a higher discounted reward compared to taking a longer path where the same reward would be less valuable due to discounting. The Q-function trained by Eq. 7 also tends to favor the shortest path. In these experiments, we simply show that BC limits the agent's performance by closely mimicking the provided suboptimal dataset. In contrast, our method allows the extraction of the best actions according to the Q-function, ignoring the suboptimal actions. We will clarify this section in the final version.
>
> **Q3. It is difficult to understand how PPL, PPL-CQL, and PPL-R work. A pseudocode for each variant would be helpful.**
>
> All of these methods mainly differ in the way they learn the Q-function (cost), not the policy extraction. We agree with the reviewer that the description of the variants is an important component that needs to be discussed. Below is the high-level pseudocode for these methods. We will add the complete pseudocode in the appendix and refer to them in main text. Please note that for ReBrac $-Q^k(s, a) +BC$ cost is using.
>
> ```
> Input: Dataset D(s,a,r,s')
> Initialize:  Q, π, f, β, α, w, ℛ
>
> ----------Update the Q-Function (cost)---------------
> for k in 1...N do
>
>     (s, a, r, s') ← sample a batch of transitions from D
>
>     if One-Step RL then pre-train Q by:
>         Q^{k+1} ← argmin_{Q} E_{(s, a, s') ~ D} [(r(s, a) + γ E_{a' ~ β(s')} [Q^k(s', a')] - Q^k(s, a))^2]
>
>     else if CQL then
>         Q^{k+1} ← argmin_{Q} E_{s ~ D, a ~ π(s)} [Q^k(s, a)] - E_{s ~ D, a ~ β(s)} [Q^k(s, a)]
>                   + (1/2) E_{(s, a, s') ~ D} [(r(s, a) + γ E_{a' ~ π(s')} [Q^k(s', a')] - Q^k(s, a))^2] + ℛ(π)
>
>     else if ReBrac then
>         Q^{k+1} ← argmin_{Q} E_{(s, a, s') ~ D} [(r(s, a) + γ E_{a' ~ π(s')} [Q^k(s', a') - α(π(s') - a')] - Q^k(s, a))^2]
>     end if
>
> ----------Update OT---------------
>
>     f^{k+1} ← argmin_f E_{s ~ D, a ~ π^k(s)} [f^k(s, a)] + w E_{s, a ~ D} [f^k(s, a)]
>     π^{k+1} ← argmin_π E_{s~D, a~π^k(s)} [-Q^k(s, a) - f^k(s, a)]
> end for
> ```
>
> **Q4. The paper has a huge room for improvement in terms of formatting. Excessive use of underlines, underfull hbox on Line 6 of Algorithm 1, and misaligned s and ragged purple boxes in the tables hurt the paper's readability.**
>
> We will incorporate your suggestions to improve the formatting. All misalignments will be corrected and the purple boxes will be replaced with underlining. In the main text, underlining will be minimized. If you have any other suggestions for improving the presentation of the paper, we would appreciate it.
>
> **Q5: Have you tried using a pre-trained Q-function learned by IQL or any other in-sample value learning algorithms instead of training the Q-function in parallel?.**
>
> No. We do not use IQL because this method does not align with the Optimal Transport framework. If we look at the OT optimization problems (Eq. 1, 4, 12), we can see that the map (policy) outputs are used as inputs for the cost function (Q-function in our case) during optimization. However, the IQL method is a weighted behavior cloning approach and not use learned policy outputs as inputs for the action-value function.
>
> **Concluding remarks:** In conclusion, we truly value your reviews. We hope that the revisions and clarifications will influence and improve your overall opinion. If we've managed to resolve your principal concerns and questions, we'd be thankful for your endorsement through an elevated score of our submission. On the other hand, if there are remaining issues or questions on your mind, we're more than willing to address them.

---

> > ### Comment · Reviewer_iRBG · 2024-08-09
> >
> > Thank you for your response. I noticed some misunderstandings and wanted to clarify a few.
> >
> > **Q1.**  If I understood the paper correctly, you are viewing $\pi\colon S\to A$ as a transport map from the state distribution $d(\cdot)$ to the behavior policy $\beta(\cdot\mid s')$ for some $s'\in S$. The transport map will depend on the choice of $s'$. How do you choose this $s'$?
> >
> > **Q2.** Lines 190 and 191 state that the agent has to "navigate through 50 intermediate steps." Doesn't this mean that the episode length is fixed to 51 regardless of the agent's path?
> >
> > **Q5.** The value learning part of IQL can be completely separated from the policy learning part, which uses AWR, as you mentioned. IQL Q function can be trained without weighted behaviour cloning and thus can be used together with the proposed algorithm.

---

> ### Author Response · Authors · 2024-08-12
>
> **Q1. If I understood the paper correctly, you are viewing  as a transport map from the state distribution d(s) to the behavior policy $\beta(\cdot|s')$ for some s'. The transport map will depend on the choice of s'. How do you choose this s'?**
>
> Sorry for the misunderstanding. We are randomly sampling $s$ from the given dataset $\mathcal{D}$, please see derivations at lines 131 and Eq.9. For the sampled $s$ we choose the conditional distribution of the action space provided by the dataset (expert) $\beta(\cdot|s)$ as the target. We will make this explicit in the final version.
>
> **Q2. Lines 190 and 191 state that the agent has to "navigate through 50 intermediate steps." Doesn't this mean that the episode length is fixed to 51 regardless of the agent's path?**
>
> Yes, the length of the episode is fixed. This length of 50, corresponds to the number of steps provided by each expert policy in the dataset.
>
> **Q5. The value learning part of IQL can be completely separated from the policy learning part, which uses AWR, as you mentioned. IQL Q function can be trained without weighted behaviour cloning and thus can be used together with the proposed algorithm.**
>
> Thanks for the clarification, this experiment is really interesting. Following your question, we conducted a series of experiments on MuJoCo using the CORL[1] IQL implementation. In these tests, we found that the value function trained using expectile regression from the IQL method is inappropriate for off-policy evaluation.
>
> - First, we conducted an experiment *without* using our method. We trained a $Q$-function using  expectile regression loss and attempted to extract a policy through direct optimization: $\min_{\pi}\mathbb{E}_{s \sim \mathcal{D}, a\sim\pi(s)} \big[-Q^{\text{IQL}}(s, a)\big]$. This approach resulted in zero rewards.
> - Second, we added a BC objective to avoid distribution shift: $\min_{\pi}\mathbb{E}_{s \sim \mathcal{D}, a\sim\pi(s)} \big[-Q^{\text{IQL}}(s, a)\big]+(a-\beta(s))^2$. The results were the same — $Q$-functions trained via expectile regression dramatically overestimate actions sampled by the learned policy $\pi$.
> - Finally, we tested our method: $\min_{\pi}\mathbb{E}_{s \sim \mathcal{D}, a\sim\pi(s)} \big[-Q^{\text{IQL}}(s, a)-f(s,a)\big]$ and observed improvements in the scores! Even with the ill-suited  cost function, policy optimization with respect to the potential $-f(s,a)$ yielded the highest scores.
>
> We also tested the advantage $A(s,a) $and exponential advantage functions exp$A(s,a)$ from IQL, but did not observe any improvements. The table below summarizes our analyses.
>
> ---
>
> **Table: Averaged normalized scores on MuJoCo tasks. Reported scores are the results of the final 10 evaluations and 3 random seeds.**
>
> | Dataset                 | $-Q^{IQL}(s, a)$ | $-Q^{IQL}(s, a)$ + BC | $-Q^{IQL}(s, a) - f(s,a)$ (Ours) |
> |-------------------------|------------------|-----------------------|---------------------------------|
> | HalfCheetah-medium             | -2.53 ± 0.1      | -2.54 ± 0.1           | 48.7 ± 0.3                     |
> | HalfCheetah-medium-expert      | -2.53 ± 0.2      | -2.54 ± 0.2           | 39.9 ± 1.4                     |
> | Hopper-medium           | 0.6 ± 0.1        | 0.6 ± 0.1             | 27.9 ± 15.4                    |
> | Hopper-medium-expert    | 0.7 ± 0.1        | 0.6 ± 0.1             | 8.7 ± 4.6                      |
> | Walker-medium           | -0.16 ± 0.1      | -0.16 ± 0.0           | 37.6 ± 3.6                     |
> | Walker-medium-expert    | -0.23 ± 0.2      | -0.21 ± 0.1           | 17.5 ± 14.3                    |
> ---
>
> While the IQL method can be formally decoupled, we see that both components, expectile-based in-sample value learning and weighted behaviour cloning, are important parts of each other to achieve strong results. Adapting IQL value learning for better off-policy evaluation is a promising direction, but beyond the scope of our contribution. We believe that our strong performance on most tasks, combined with different types of a cost function, justifies our formulation and makes it valuable to the community. We will include these analyses in the final version of the paper.
>
> **Reference:**
>
> [1] JAX-CORL: https://github.com/nissymori/JAX-CORL

---

> > ### Comment · Reviewer_iRBG · 2024-08-12
> >
> > Thank you for your response.
> >
> > **Q1.** I think I've failed to deliver my point. A transport map between the state distribution $d$ and the behaviour policy $\beta(\cdot\mid s)$ for some $s$ depends on the choice of $s$. This means the optimal transport framework will produce multiple policies $\pi_s$, one for each $s\in S$. How are you going to combine them into one policy $\pi$? It seems like you are defining $\pi(s)=\pi_s(s)$ for each $s$ (am I correct?), but this does not make much sense because any change on a measure-zero set {$s$} will not affect the result. Therefore, any arbitrary function $\pi$ would be a solution, which is definitely not what we want.
> >
> > **Q2.** If the episode length is fixed to 50, the discounted return will always be $\gamma^{50}$ regardless of the agent's policy.
> >
> > **Q5.** Thank you for running the experiments. It is interesting that using IQL Q-functions as the cost function fails so miserably.

---

> > > ### Author Response · Authors · 2024-08-13
> > >
> > > **Q1.** We learn a single neural network $\pi_\theta$ that approximates all policies $\pi_\theta(s) \leq w(\beta(\cdot|s))$ for each state $s$. But the same is true for any BC method in offline RL when we learn $\pi_\theta(s)=\beta(\cdot|s)$ using $\ell^2$, KL, or any other discrepancy. The neural approximator for the policy $\pi$ is able to generalize and provide a solution for the measure-zero set of the $d^\beta(s)$. Did we understand your question correctly?
> > >
> > > **Q2.**  Yes, you are right, this was an oversight on our part. Only this toy experiment was affected. We thank the reviewer for pointing this out. We fixed this by providing the expert trajectories with a different length in the dataset. This gave us the same visual results.
> > >
> > > P.s. Although there were no explicit signals with fixed episode length, the model still learned the near shortest path. We think this is because the $Q$ function approximation implicitly learned to value actions similar to the final rewarded action more highly.
> > >
> > > **Q5.** Yes, it was really interesting results. We will make these experiments public along with the rest of the code.

---

> ### Comment · Reviewer_iRBG · 2024-08-13
>
> **Q1.** Say we are going to use the log-likelihood as our regularizer, which means the objective function would be something like $\mathbb{E}_{s\sim \mathcal{D}}\left[Q(s, \pi(s))-\alpha\log\beta(\pi(s)\mid s)\right]$. For each $s\in S$, the regularizer $\log\beta(\pi(s)\mid s)$ is only affected by the value of $\pi(s)$ and nothing else. If we ignore the expressivity of a neural network, the resulting $\pi$ is straightforward to interpret: $\pi(s)=\text{arg}\max_a Q(s, a)-\alpha\log\beta(a\mid s)$. However, in the case of OT, the value of $T_s(s')$ for all $s'\in S$ matters, where $T_s$ is the transport map between the state distribution $d(\cdot)$ and $\beta(\cdot\mid s)$. I'm concerned that trying to force the same $T$ for all $\beta(\cdot\mid s)$ would cause conflicts between different $s$.
>
> **Q2.** Does it mean that an agent's actions in the intermediate 49 steps wouldn't affect the return?

---

> > ### Author Response · Authors · 2024-08-13
> >
> > **Q1.** Thank you for the clarification. Optimal transport does not cause conflicts between different $s$. Our final objective (Eq.12) for policy is similar to the BC example that your gave: $\min_\pi\mathbb{E}_{s \sim \mathcal{D}} [-Q(s, \pi(s))-f(s,\pi(s))]$. For each $s$ the potential $f$ is affected by the value of $\pi(s)$ and nothing else.
> > In the optimal transport literature, several methods with a similar objective have been considered to map into the conditional distributions [1, Eq. 8c][2, Sec. 6.2]. Please note that [1] considered the case when even $T$ is conditional, and then, use the single $T$ for all distributions, see Sec. 2.4.
> >
> > **Q2.** Yes, intermediate steps are not rewarded, but the $Q$-function is particularly given to estimate these intermediate steps. We added different lengths to the dataset, 20, 30, 50. Consequently, trajectories that lead to the reward faster have a higher $Q$- value.
> >
> > For a simpler illustration, we also made the reward for each step equal to the Euclidean distance between the current state-action pair and the final rewarded state-action pair. This simplifies the learning problem, but gives a clear intuition why the shortest path is optimal. If you find this more relevant, we will include such an experiment in the final version.
> >
> >
> > **Reference:**
> >
> > [1] Nonlinear Filtering with Brenier Optimal Transport Maps: https://openreview.net/pdf/70633c38d3ce64c9b3b29dd7abd18c2f6b6e1dc6.pdf
> >
> > [2] Neural Monge Map estimation and its applications: https://openreview.net/pdf?id=2mZSlQscj3

---

> ### Comment · Reviewer_iRBG · 2024-08-14
>
> **Q1.** I'm confused now. Let's consider the simplest case where $\mathcal{D}$ has two elements $(s_1, a_1)$ and $(s_2, a_2)$. Then the state distribution $d(s)$ is $\frac{1}{2}\delta(s=s_1)+\frac{1}{2}\delta(s=s_2)$ and the behavior policies are $\beta(a\mid s_1)=\delta(a=a_1)$ and $\beta(a\mid s_2)=\delta(a=a_2)$. The optimal transport map $T_{s_1}$ from $d$ to $\beta(\cdot\mid s_1)$ should be $T_{s_1}\equiv a_1$ and the map $T_{s_2}$ from $d$ to $\beta(\cdot\mid s_2)$ should be $T_{s_2}\equiv a_2$. If you try to use the same $T$ in both cases, conflict would occur since $a_1\neq a_2$, wouldn't it? Am I missing something?
>
> **Q2.** What is the transition function? From the description in the paper, it looks like there are states $s_1, s_2, \cdot, s_T$ uniformly spaced across the x-axis and the agent will transition from $s_i$ to $s_{i+1}$ regardless of the action it takes. After 50 steps, the agent will arrive at state $s_T$ no matter what and get a reward of 1. What would encourage the agents to follow the shortest path?

---

> > ### Author Response · Authors · 2024-08-14
> >
> > **Q1:** Dear reviewer. As we already noted, we consider the conditional optimal transport setup. This means that we want to simultaneously learn a family of transport plans (indexed by some condition $c$; each plan is between some distributions $p_c, q_c$). Following the standard practices of neural OT, this would means that we have to learn a map $T(c, x, z)$, where $c$ is a condition, $x$ is an input point (from $p_c$) and $z$ is a random noise to be able to learn stochastic plans.
> >
> > In our case, we need to learn a set of OT plans, each plan is conditioned on the given state $c=s$. Following our problem, each such plan should be a plan between distribution $p_c=\delta_s$ (this is our choice) and $q_c=\beta(\cdot |s)$. Hence, we would have to learn a function $T(c,x,z)=T(s,s',z)$ with 3 arguments. However, since the input $s'$ comes from $\delta_s$, it always coincides with $s$ with probability 1. Hence, one may merge the arguments together to simply get a function of the form $T(s,z)$. In turn, we also remove the random noise component $z$, which is standard practice in neural OT methods, unless additional regularization (variance, entropy, etc.) is used[1]. Hence, our final function is $T(s)$. We will add this details into the final version.
> >
> > **Q2:** The transition function is $P(s'|s,a)=1$. As we said in the previous answer, we have redefined the environment to avoid any confusion. Now the reward is given for each state-action pair $r(s_t,a_t) = -\ell^2((s_t,a_t), (S_T, A_T))$, where $S_T=0$ and $A_T=0.2$ is the target state-action pair. In this scenario, trajectories deviating from the straight line (actions define y-axis coordinates) from $S_0$ to $S_T$ will have the lower cumulative reward $\sum^{T}_0\gamma^t r(s_t,a_t)$.
> >
> > **References**
> > [1] Neural Optimal Transport: https://openreview.net/pdf?id=d8CBRlWNkqH

---

> > > ### Comment · Reviewer_iRBG · 2024-08-14
> > >
> > > Now I get it. So by the *state distribution* $d(s)$ in line 129, you meant the Dirac mass $\delta(s)$. Please explicitly mention it in the paper. *State distribution*, especially when written as $d^\beta$, often means the state occupancy measure under policy $\beta$, so I highly recommend you use another term to prevent misunderstandings like mine. As all of my concerns were resolved, I have updated my score accordingly.

---

### Official Review · Reviewer_qCSm · 2024-07-14

**Soundness:** 3
**Presentation:** 2
**Contribution:** 2
**Rating:** 6
**Confidence:** 4

**Summary:**

The authors address the problem of offline RL. They rethink offline RL using optimal transport. In offline RL, often the datasets consist of several sub-optimal trajectories that are needed to be stitched together. The authors use partial OT to incorporate stitching and a maxmin formulation of this partial OT. They perform experiments majorly on D4RL.

**Strengths:**

(1) While a lot of methods that use OT, use Wasserstein distance and that requires optimizing a function constraint to be Lipschitz. This is often hard. The authors have used a maxmin formulation which does not need the function to be Lipschitz.

(2) They treat offline RL as an OT problem rather than using OT as a regularizer.

**Weaknesses:**

(1) There should have been comparisons to W-BRAC.

(2) The results show that PPL^{CQL} produces some marginal improvement over PPL and PPL^{R} with ReBRAC.

(3) From Equation 12, you must not need \beta. But in experiments you constantly talk about being in conjugation with something or training a \beta. Maybe this wasn't clear or I misunderstood, why do you need to be in conjugation with CQL or ReBRAC or one-step RL? Why can't you simply train the maxmin objective in equation 12.

The work is interesting but I would like the authors to clarify why at all there is a need for conjugation? IQL performs better than CQL, so why would I use PPL^{CQL} and not train IQL directly? I request the authors to clarify the contribution of this paper in context to this.

**Questions:**

(1) What is the motivation for using OT here?

(2) what is d^\beta(s). Is it the visitation of the behavioral policy \beta? Then why do you say that you want to learn a policy that transfers mass from d^\beta(s) to the corresponding distribution given by the behavioral policy?

**Limitations:**

Limitations are addressed.

---

> ### Author Rebuttal · Authors · 2024-08-07
>
> Thank you for your thoughtful questions. We have managed to address and improve the paper based on them. Below, we address each of your concerns: we include the W-BRAC comparison, provide an explanation of the conjunction, clarify the OT motivation, and provided details on the behavior policy visitation.
>
> **Q1: There should have been comparisons to W-BRAC.**
>
> We will include W-BRAC's results in Table 2 of our paper. Initially we compared our method with other methods that have already shown significant improvements over W-BRAC, suggesting a clear performance improvement hierarchy.
>
>
>
> **Q2: ... Why can't you simply train the maxmin objective in equation 12.? ... Why at all there is a need for conjugation  with CQL or ReBRAC or one-step RL? IQL performs better than CQL, so why would I use $\text{PPL}^{CQL}$ and not train IQL directly? I request the authors to clarify the contribution of this paper in context to this.**
>
> To address the comment regarding the conjunction, we would like to clarify several reasons for that:
>
> - We don't simply train Eq. (12), because the Q-function trained via Eq. (7) and then used in Eq. (12) can suffer from overestimation bias (lines 107-113).
>
> - Consequently, we tested our method in conjunction with various methods that avoid overestimation of Q-function. $\beta$ is actually necessary for the methods used. Please note that the contribution of our method is policy extraction, not solving the overestimation bias. These overestimation-avoiding methods are used to obtain different types of cost functions in our method, showing that our method can work efficiently with any of them.
>
> - We do not use IQL as a backbone because this method does not align with the Optimal Transport framework. If we examine the optimization problems (Eq. 1, 4, 12), we can see that the map (policy) outputs are used as inputs for the cost function (Q-function in our case) during optimization. However, the IQL method is a weighted behavior cloning approach, which does not use policy outputs as inputs for the critic function. Instead, only actions from the dataset are weighted by the action-value function.
>
>
> In summary, we clarify our contribution: We proposed a novel optimal transport-based policy extraction method and provided an analysis of its performance on various RL-based cost functions. We will add this very explicitly to the final version of the paper.
>
>
> **Q3: The results show that $\text{PPL}^{CQL}$ produces some marginal improvement over PPL and $\text{PPL}^{R}$ with ReBRAC.**
>
> The goal of these experiments was to show a side-by-side comparison between the basic method and its improved version via our method. In Table 1, we compare $\text{PPL}^{CQL}$ to CQL and $\text{PPL}^{R}$ to ReBRAC, showing that regardless of the backbone used, our method allows improvement.
>
>
> **Q4: What is the motivation for using OT here?**
>
> This motivation follows from the need to have a *partial* policy that maps only to the best action distribution when dealing with suboptimal data in offline learning. In OT, *partial* alignment methods have been extensively developed. To integrate OT methods into RL in the simplest way, we considered offline RL as an *max-min* OT problem, meanwhile avoiding the limitations of existing OT in RL methods.
>
> **Q5: What is $d^\beta(s)$. Is it the visitation of the behavioral policy $\beta$? Then why do you say that you want to learn a policy that transfers mass from $d^\beta(s)$ to the corresponding distribution given by the behavioral policy?**
>
> Yes, you are right. Indeed, it is the visitations of the behavioral policy. Please note that our method is completely offline. Thus, the probability over the states, $d^\beta(s)$, is the only distribution over the state space that we have. We have no ability to get our distribution, $d^\pi(s)$, as we cannot interact with the environment. This is the standard way for offline RL, not something we came up with. For each state visited by the expert, we aim to map it to the most efficient part of the distribution over the actions for this state, provided by $\beta(\cdot|s)$.
>
>
> **Concluding remarks:** We truly value your reviews. Please respond to us, to let us know if the clarifications above, suitably address your concerns. If you finds the responses above are sufficient, we kindly ask that you consider raising score.

---

> > ### Comment · Reviewer_qCSm · 2024-08-11
> >
> > Thank you for clarifying my doubts. I have increased my score.

---

> ### Comment · Area_Chair_ZGRe · 2024-08-11
> **Please respond to authors**
>
> Hello reviewer qCSm: The authors have responded to your comments. I would expect you to respond in kind.

---

### Decision · Program_Chairs · 2024-09-25

**Decision:**

Accept (poster)

**Comment:**

The paper presents an approach to offline reinforcement learning based on multiple different (potentially suboptimal) expert data. The problem is then to "stitch" together a policy from the best actions across experts. The authors formalize the problem as partial Optimal Transport. This is a novel, interestingly different role for Optimal Transport than has appeared in previous work on reinforcement learning. The results show promising performance against various baselines.

The authors are in consensus that this is interesting and sound work. I have read the reviews and the paper and agree. This is novel work that will likely be of broad interest to the community.